# Behavioural risk factors for cardiovascular diseases among adolescents of secondary school in Tulsipur Sub-Metropolitan City, Nepal: A cross-sectional study

Sita Bista[1]*, Bishow Puri[1], Sanju Maharjan[1], Poshan Thapa[2‡], Buna Bhandari[3,4‡]

**1** Central Department of Public Health, Tribhuvan University Institute of Medicine, Kathmandu, Nepal, **2** School of Population and Global Health, McGill University, Quebec, Canada, **3** Florida State University College of Nursing, Tallahassee, Florida, United States of America, **4** Department of Global and Population Health, Harvard Chan School of Public Health, United States of America

‡ Shared senior authors
* sitabista507@gmail.com

## Abstract

### Background

Cardiovascular diseases (CVDs) are a leading cause of global death and disability, affecting one-third of adult population. Often overlooked in school-going adolescents, behavioural risk factors are crucial contributors to CVD risk which begin early and accelerate during adolescent period. This study aims to assess the behavioural risk factors and their associated determinants among adolescents of Tulsipur Sub-Metropolitan City, Nepal.

### Methods

A school-based cross-sectional study was conducted among 361 adolescents aged 16–19 years studying in grade 11 and 12 from public and private schools. Schools were selected using a stratified proportionate sampling method. Data were collected through a self-administered, structured, and validated questionnaire covering socio-demographic characteristics, behavioural risk factors of CVDs, and parental information. Descriptive and analytical statistics were used to analyse the data.

### Results

The most prevalent behavioural risk factor was the consumption of calorie drinks (99%), followed by sedentary behaviour (60%), insufficient fruit and vegetable intake (57%), physical inactivity (35%), and consumption of processed food high in salt (33%). The prevalence of current smoking, alcohol consumption, and smokeless tobacco use was 12%, 10%, and 9% respectively. Key factors associated with the behavioural risk include maternal education, ethnicity, and education system.

**Data availability statement:** All relevant data are within the manuscript and its Supporting Information files.

**Funding:** The author(s) received no specific funding for this work.

**Competing interests:** The authors have declared that no competing interests exist.

Parental tobacco and alcohol use were also associated with adolescent smoking and alcohol consumption.

## Conclusions

The high prevalence of CVD risk factors among adolescents in Nepal highlights the urgent need for targeted interventions in both household and school settings. These interventions should aim to reduce behavioural risk factors to prevent the future burden of CVDs in resource-limited areas like Nepal.

---

### Introduction

Cardiovascular disease (CVD) is the leading cause of preventable death worldwide, responsible for 17.9 million deaths in 2019, which represented 32% of all global deaths [1]. If current trend continues, it is estimated that approximately 23 million individuals will die from CVD by 2030 [2]. CVD affects one-third of adult population globally, making it a growing epidemic and one of the most significant public health challenges [3]. This is particularly concerning for adolescents aged 10–24 years, who represent over 25% of the global population and are increasingly at risk of developing CVD in adulthood [3]. The burden of CVD is particularly high in low- and middle-income countries (LMICs) including in South Asia, where three-quarters of these deaths occur [1].

In Nepal, CVD contributed to 26.9% of all deaths and 12.8% of total Disability-Adjusted Life Years (DALYs) in 2017 [4]. Of particular concern is the mortality rate among adolescents, with CVD-related deaths in the 15–19 age group reported at 3.9 per 100,000, accounting for 4.8% of total CVD deaths in the country [4]. This early onset of CVD risk in adolescence not only has direct health implications but can also place a heavy burden on Nepal's macroeconomic conditions, as CVD typically affects individuals in their most productive years [1].

Evidence shows that physical inactivity, unhealthy diets, smoking, and alcohol consumption are major CVD risk factors among adolescents [3]. These behaviours often begin in childhood and become more pronounced during adolescence, though the clinical manifestations of CVD generally appear in adulthood [2]. Many LMICs, including Nepal, are experiencing rapid, unplanned urbanization, which has led to reduced physical activity spaces for children. Simultaneously, increased access to junk food and fast food has made adolescents more vulnerable to the development of CVD risk factors [5].

Despite these growing concerns, limited research exists on the behavioural risk factors for CVD in semi-urban areas of Nepal especially among understudied adolescent population. This study seeks to address that gap by assessing the prevalence of CVD risk and its associated factors. The findings aim to inform the development of adolescent-focused public health interventions that are both practical and effective in resource-limited settings like Nepal, with a focus on preventing the escalating burden of CVD.

## Methods

### Study design and setting

This cross-sectional study was conducted among secondary-level school-going adolescents aged 16–19 in Tulsipur Sub-Metropolitan City, Nepal. Tulsipur is located in the mid-western region of Nepal, a rapidly urbanizing area where people from neighbouring districts relocate for better education and opportunities. The city's recent growth, including the emergence of fast-food chains and changing lifestyles, poses an increased risk of CVDs (6).

### Study period

The study was conducted in 6 months from November 2021 to May 2022, from inception of research through literature review and problem identification to final data analysis and report preparation. The data was collected from 25th March to 8th April 2022.

### Study population

The population consisted of students in grades 11 and 12 from public and private schools. Eligible participants were students between the ages of 16 and 19 years. Students with a history of CVDs or other chronic diseases were excluded from the study.

### Operational Definition

Operational definitions used in the study are provided in S1 Appendix.

### Sample size

The total calculated sample size was 367 students, based on the following assumptions: the total number of secondary-level school adolescents (16–19 aged) in Tulsipur Sub-Metropolitan City (N) = 5,262, a confidence level of $\alpha = 0.05$, $Z = 1.96$ (for 95% confidence interval), $p = 0.366$ (prevalence of tobacco use according to the STEPs survey in Lumbini Province, 2019 (7)), and a 10% non-response rate.

### Sampling

A stratified proportionate sampling method was employed using the sampling frame obtained from municipal education section. The two strata were defined based on schools' affiliation (Public/Private), with the sample size proportionally distributed based on the student population in each stratum. Two public and two private secondary schools were randomly selected. Within each school, students were selected through a simple random lottery method using a random number generator. The students were divided into grades 11 and 12, with a proportional distribution according to age. The student's records and details of each selected school were obtained from the school administration. Each student was assigned a unique identification number. Based on proportionate sample of selected schools, Microsoft excel was used to generate a predefined and non-repeating number. The randomly generated numbers were cross-verified with the student details. Thereafter, selected students were invited to participate in the study. If selected students did not provide consent or were absent, alternative students were taken through additional random numbers generated. The sampling method is illustrated in Fig 1.

### Data collection tools and techniques

Data were collected using a self-administered questionnaire developed in Nepali after obtaining written consent from parents and verbal consent from students following ethical guidelines. The structured questionnaire was adapted from a study by Islam et al., conducted among school-going children in Bangladesh, with permission from the original authors [6].

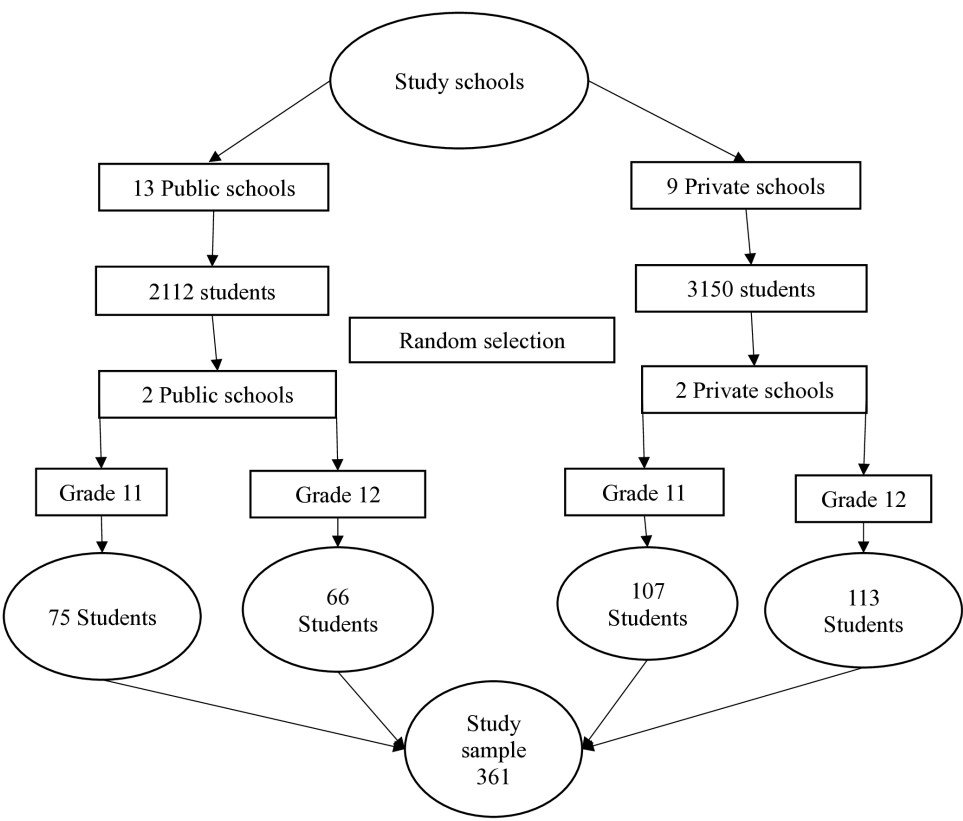

**Fig 1. Sampling method used in recruiting participants.**

The questionnaire contained 18 screening questions covering the five primary CVD risk behaviours. It was pretested with 10% of the sample size, i.e., among 37 students from Ghorahi Sub-Metropolitan City, a location with similar characteristics to the study area and revised as necessary. The pre-testing data was not included in final analysis. Data was collected by first and second authors (SB and BP) and regular supervision was maintained to minimize information bias, ensuring no more than 15 participants were in a room at any time. Any questions or concerns were addressed before the data collection process began.

## Data processing and analysis

All data were systematically compiled, coded, checked, and edited on the same collection day. Data were entered into EpiData version 4.6, rechecked, and cleaned to ensure data quality. Statistical analysis was conducted using SPSS version 22.0. Univariate analysis was used to calculate frequencies, percentages, means, medians and standard deviation for the socio-demographic variables and parents' information. Chi-square analysis was used to assess the association between socio-demographic characteristics, parental information, and the behavioural risk factors of CVDs. Variables with p<0.25 from chi-square were considered for univariate and multivariate logistic regression analysis. Confounding was assessed using a threshold of 20% change, and all variables were retained in the final adjusted model to control for confounding.

Ethical considerations: Ethical approval was obtained from the Institutional Review Committee of Tribhuvan University Institute of Medicine, Nepal [IRC no: 355(6–11) E2 078/079]. Both written parental consent and the student's consent

were obtained for students under 18. On the first day, Parents received a parental consent form through students. Only after obtaining parental consent, students were given a written consent form. Students aged 18 or above provided their written informed consent. Anonymity and confidentiality were strictly maintained by not disclosing any personally identifiable information.

## Results

The self-administered questionnaire was completed by 367 participants, but six responses were incomplete. Thus, the final analysis included 361 responses.

### Characteristics of the participants and parents

The mean age of the participants was 17.5±0.92 years. More than half (54%) were female, and 70% belonged to the Brahmin/Chhetri ethnicity. Around half (50.4%) were in grades 11, and 61% were enrolled in the management faculty. Most participants (66.8%) attended private schools, and 76% lived with family members.

About one-quarter (24%) of the respondents' mothers and 10.5% of their fathers had no formal education. The majority of fathers (29%) were involved in business, while 63% of mothers were homemakers. In total, 20% of respondents reported that their parents had chronic illnesses, 15% responded their parents smoked, 24% indicated smokeless tobacco use, and 27% reported alcohol consumption. Additional information can be found in Table 1.

### Distribution of CVD risk factors among the participants

The most prevalent behavioural risk factor was the consumption of calorie-dense drinks (99%), followed by sedentary behaviour (60%) and insufficient fruit and vegetable intake (57%) was the third most common risk factor. These findings are presented in Fig 2.

### Association between CVD Risk Factors and Socio-demographic Characteristics

Chi square test revealed processed food high in salt consumption was associated with grade (p=0.007), ethnicity (p=0.028), and mother's education (p=0.012). Added salt intake was also associated with the mother's education (p=0.049). However, calorie drink consumption did not show any significant associations. Detailed results are provided in Table 2.

Further analysis showed that sedentary behaviour was associated with the education system (p=0.003), and physical inactivity was associated with the education system (p<0.001), mother's education (p=0.025), and father's education (p=0.046). Insufficient fruit and vegetable intake did not show significant associations with any variables. These findings are presented in Table 3.

Similarly, the use of refined vegetable oil was significantly associated with ethnicity (p=0.014), education system (p=0.001), current living conditions (p=0.026), parental chronic illness (p=0.026), and parental alcohol use (p=0.003). Current smoking was associated with gender (p =<0.001), ethnicity (p=0.012), mother's education (p=0.029), parents' smoking habits (p=0.002), and parents' smokeless tobacco use (p =<0.001). Additionally, current alcohol use was significantly associated with gender (p =<0.001), parents' smokeless tobacco use (p=0.017), and parents' alcohol use (p=0.022). Current smokeless tobacco use was significantly associated with gender (p =<0.001). These results are shown in Table 4.

The logistic regression model findings showed that processed food high in salt consumption was significantly associated with sex, ethnicity, grade and mother's education. However, calorie drink consumption and added salt intake did not show any significant associations. Sedentary behaviour was significantly associated with education system, parents' tobacco use and parent's alcohol use. Similarly, insufficient fruit and vegetable intake and physical activity was associated

**Table 1. Social and demographic characteristics of participants and Parents (n = 361).**

| SOCIO-DEMOGRAPHIC CHARACTERISTICS OF PARTICIPANTS | NUMBER | PERCENTAGE |
|---|---|---|
| **AGE** | | |
| Mean ± SD | 17.5 ± 0.92 | |
| **GENDER** | | |
| Male | 164 | 45.4 |
| Female | 195 | 54.0 |
| Do nott want to disclose | 2 | 0.6 |
| **ETHNICITY** | | |
| Brahmin/Chhetri | 255 | 70.6 |
| Janajati | 68 | 18.8 |
| Dalit/Muslim/others | 68 | 10.6 |
| **RELIGION** | | |
| Hindu | 354 | 98.1 |
| Others | 7 | 1.9 |
| **GRADE** | | |
| 11 | 182 | 50.4 |
| 12 | 179 | 49.6 |
| **EDUCATION SYSTEM** | | |
| Private | 241 | 66.8 |
| Public | 120 | 33.2 |
| **CURRENT LIVING CONDITION** | | |
| Living with family members | 274 | 76 |
| Renting a room with friend | 61 | 16.9 |
| Alone/others | 26 | 7.2 |
| **CHARACTERISTICS OF PARENTS** | | |
| **Mother education** | **n = 357\*** | |
| No formal schooling | 90 | 25.2 |
| Primary school and below | 146 | 40.9 |
| Secondary school | 60 | 16.8 |
| High School or above | 41 | 16.9 |
| **Mother occupation** | **n = 357\*** | |
| Unemployed | 233 | 65.3 |
| Agriculture | 47 | 13.2 |
| Business | 39 | 10.9 |
| Others¶ | 38 | 10.6 |
| **Father education** | **n = 350#** | |
| No formal education | 37 | 10.6 |
| Primary School or below | 106 | 30.3 |
| Secondary school | 101 | 28.9 |
| High School or above | 106 | 30.3 |
| **Father occupation** | **n = 350#** | |
| Unemployed | 27 | 7.7 |
| Agriculture | 101 | 28.9 |
| Business | 105 | 30 |
| Others¶ | 117 | 33.4 |
| **Chronic illness of father or mother or both** | **n = 358§** | *multiple response |
| Hypertension | 48 | 13.4 |

*(Continued)*

**Table 1.** (Continued)

| SOCIO-DEMOGRAPHIC CHARACTERISTICS OF PARTICIPANTS | NUMBER | PERCENTAGE |
|---|---|---|
| Heart Disease | 10 | 2.8 |
| Diabetes mellitus | 11 | 3 |
| Cancer | 1 | 0.3 |
| Chronic respiratory | 4 | 1.1 |
| **Behavioral risks of father or mother or both** | **n=358§** | *multiple response |
| Smoking habit | 53 | 14.8 |
| Smokeless tobacco | 88 | 24.4 |
| Alcohol use | 98 | 27.4 |

*4 children do not have mother*

#*11 children do not have father*

§*3 children neither have father nor mother*

Others¶ *include foreign employment, government employee, non-government employee and retired from government job.*

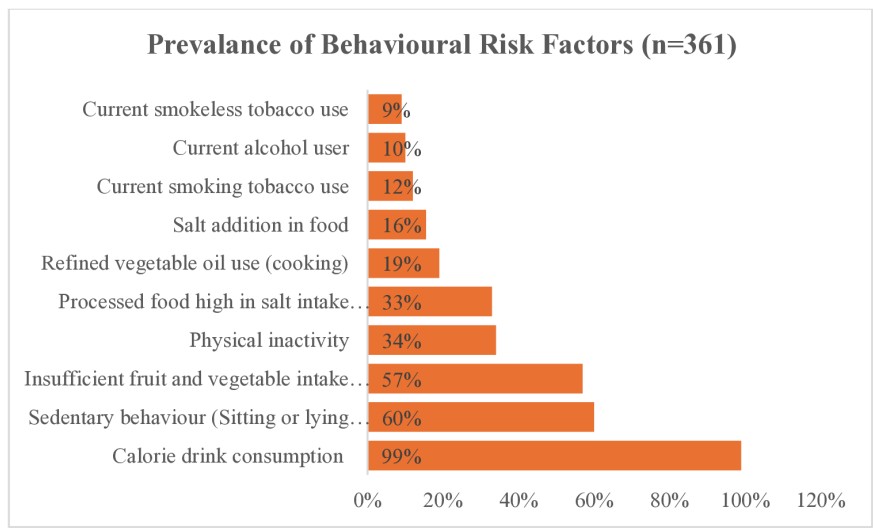

**Fig 2. Prevalence of behavioral risk factors among secondary school adolescents.**

with education system only. Additional findings are presented in Table 5 and refer to S1 Table for further details of regression analysis.

## Discussion

The findings of this study highlight a significant burden of behavioural risk factors for CVD among school-going adolescents in Nepal. The study found a high prevalence of consumption of calorie-dense drinks (99%), sedentary behaviour (60%), low fruit and vegetable intake (57%), and physical inactivity (35%). Key socio-demographic factors, such as gender, ethnicity, religion and parental behaviours, were associated with these risk factors. Additionally, processed food consumption, smoking, and alcohol use were closely linked with parental habits, highlighting the strong influence of family environment on adolescent health behaviours. Our study's findings on physical inactivity (34.9%) significantly associated with education system (OR=2.810) were notably higher than those reported in the 2019 STEPS survey (10.8%) and a

**Table 2. Association of processed food high in salt consumption, calorie drink consumption and added salt intake with socio-demographic characteristics (n=361).**

| Characteristics | Categories | Processed food consumption (always or often Yes=119) n (%) | Calorie drink consumption (yes=356) n (%) | Added salt intake (always or often n=56) n (%) |
|---|---|---|---|---|
| Age | 16–17 | 63(52.9%) | 170(47.8%) | 26(46.4%) |
| | 18–19 | 56(47.1%) | 186(52.2%) | 30(53.6%) |
| | p-value | 0.181 | 0.925 | 0.808 |
| Gender[+] | Male | 47(39.5%) | 164(46.3%) | 25(44.6%) |
| | Female | 72(60.5%) | 190(53.7%) | 31(55.4%) |
| | p-value | 0.098 | 0.107 | 0.865 |
| Ethnicity | Brahmin/Chhetri | 93(78.1%) | 250(70.2%) | 40(71.4%) |
| | Others | 26(21.9%) | 106(29.8%) | 16(28.6%) |
| | p-value | **0.028**[*] | 0.327 | 0.887 |
| Religion | Hindu | 116(97.5%) | 349(98%) | 54(96.4%) |
| | Others | 3(2.5%) | 7(2%) | 2(3.6%) |
| | p-value | 0.876 | 1 | 0.662 |
| Grade | 11 | 72(60.5%) | 177(49.7%) | 33(58.9%) |
| | 12 | 47(39.5%) | 179(50.3%) | 23(41.1%) |
| | p-value | **0.007**[*] | 0.075 | 0.166 |
| Education System | Public | 43(36.1%) | 119(33.4%) | 21(37.5%) |
| | Private | 76(63.9%) | 237(66.6%) | 35(62.5%) |
| | p-value | 0.413 | 1 | 0.462 |
| Current living condition | Living with family members | 87 (73.1%) | 271(76.1%) | 43(76.8%) |
| | Others | 32(26.9%) | 85(23.9%) | 13(23.2%) |
| | p-value | 0.385 | 0.403 | 0.866 |
| Mother's[*] education | No formal schooling | 20(16.9%) | 89(25.2%) | 20(35.7%) |
| | Formal schooling | 98(83.1%) | 264(74.8%) | 36(64.3%) |
| | p-value | **0.012**[*] | 1 | **0.049**[*] |
| Mother's* occupation | Employed | 41(34.7%) | 123(34.8%) | 21(37.5%) |
| | Unemployed | 77(65.3%) | 230(65.2%) | 35(62.5%) |
| | p-value | 0.935 | 0.353 | 0.601 |
| Father's[#] education | No formal schooling | 13(11%) | 38(11%) | 7(12.7%) |
| | Formal schooling | 105(89%) | 308(89%) | 48(82.3%) |
| | p-value | 0.945 | 1 | 0.627 |
| Father's[#] occupation | Employed | 114(96.6%) | 330(95.4%) | 51(92.7%) |
| | Unemployed | 4(3.4%) | 16(4.6%) | 4(7.3%) |
| | p-value | 0.450 | 1 | 0.488 |
| Parents'[§] chronic illness | Yes | 22(18.6%) | 69(19.5%) | 8(14.3%) |
| | No | 96(81.4%) | 285(80.5%) | 48(85.7%) |
| | p-value | 0.761 | 0.583 | 0.279 |
| Parent's[§] smoking habit | Yes | 15(12.8%) | 51(14.4%) | 11(19.6%) |
| | No | 102(87.2%) | 303(85.6%) | 45(80.4%) |
| | p-value | 0.461 | 0.106 | 0.267 |
| Parent's[§] smokeless tobacco use | Yes | 31(26.5%) | 86(24.3%) | 16(28.6%) |
| | No | 86(73.5%) | 268(75.7%) | 40(71.4%) |
| | p-value | 0.558 | 0.254 | 0.450 |

*(Continued)*

**Table 2.** (Continued)

| Characteristics | Categories | Processed food consumption (always or often Yes=119) n (%) | Calorie drink consumption (yes=356) n (%) | Added salt intake (always or often n=56) n (%) |
|---|---|---|---|---|
| Parent's§ drinking habit | Yes | 34(29.1%) | 96(27.1%) | 16(28.6%) |
| | No | 83(70.9%) | 258(72.9%) | 40(71.4%) |
| | p-value | 0.618 | 0.302 | 0.827 |

+n=359 (2 children did not disclose their gender) *n=357 (four children did not have mother)

#n=350 (eleven children did not have father) §n=358(three children did not have both of parents)

*Statistically significant(p<0.05)

Percentages represent column% among those who reported the behaviour.

study from Bangladesh (8%) [6–8]. However, they were comparable to findings from Western Nepal (38%) [9] and lower than the 85% reported in the Global School-Based Student Health Survey (GSHS) [10]. These differences may reflect Tulsipur's recent urbanization and variations in socio-cultural contexts. Greater involvement in household chores and walking to school may also explain the relatively higher activity levels among our respondents.

Sedentary behaviour, reported by 60% of adolescents, was significantly associated with the education system (OR=0.351), a prevalence similar to that found in a Brazilian study (58.1%) [11]. Public schooling and lack of formal parental education often reflect lower socio-economic status, which may limit opportunities for physical activity. In aadition, urbanization, lack of open spaces for exercise, and increased screen time also contribute to sedentary behaviour among adolescents, as observed in other urban settings [12–14].

Low fruit and vegetable intake was reported by 57% of adolescents and was significantly associated with education system (OR=0.519). This figure is consistent with findings from Western Nepal (58%) [9] but lower than the 98% reported in Bangladesh [6]. These variations may be explained by differences in study populations, as our study focused primarily on urban adolescents.

Consumption of calorie-dense drinks was very high (99%), comparable to findings from Western Nepal (83.1%) [9] but considerably higher than those reported by the GSHS (33.3%) and in India (44.8%) [10,15]. This pattern may reflect the rapid urbanization of Tulsipur, where healthier food options have been increasingly replaced by fast food and sugary drinks. Despite the high prevalence, no significant associations were observed between calorie drink consumption and socio-demographic factors in this study.

The prevalence of added salt intake (15.5%) was lower than that reported in Bangladesh (66.9%) [6], likely due to differences in food practices. Processed food consumption reported by 33% of adolescents was significantly associated with gender (OR=0.597), mother's education (OR=2.895) and ethnicity (OR=2.171). This prevalence is lower than the 75% reported in the GSHS and the 49.2% observed in India [7,10,13–15]. The variation may be explained by differences in measurement, as our study focused on daily or frequent consumption. Adolescents from Brahmin/Chhetri, a comparatively advantaged group may have greater access to income and urban food systems, leading to increased processed food consumption. In addition, female adolescents may be more likely to prefer or use processed foods during food preparation. Furthermore, adolescents whose mother had no formal education may be less exposed to healthy dietary practices and thus rely more on processed foods as a part of their school snacks. Smoking rates (12%) were higher than those reported in the GSHS and in a study from Bangladesh [6,10,14]. Smoking was significantly associated with gender (OR=11.336), mother education (OR=4.675), parent's smoking (OR=3.055) and smokeless tobacco use (OR=2.745). Adolescents may imitate their parents' behaviours, as suggested by the strong association between parental and adolescent smoking. The link between smoking and lower socio-economic status, indicated by mothers' lack of formal education, is consistent with findings from previous research [15–17].

 

**Table 3. Association of sedentary behaviour, insufficient fruit and vegetable intake, and physical inactivity (n = 361).**

| Characteristics | Categories | Sedentary behaviour (yes = 217) | Insufficient fruit and vegetable intake(yes = 206) | Physical inactivity(yes = 126) |
|---|---|---|---|---|
| Age | 16-17 | 104(49.7%) | 96(46.6%) | 54(42.9%) |
| | 18-19 | 113(52.1%) | 110(53.4%) | 72(57.1%) |
| | p-value | 0.99 | 0.563 | 0.158 |
| Gender+ | Male | 98(45.6%) | 95(46.3%) | 62(50%) |
| | Female | 117(54.4%) | 110(53.7%) | 62(50%) |
| | p-value | 0.720 | 0.928 | 0.066 |
| Ethnicity | Brahmin/Chhetri | 156(71.9%) | 144(69.9%) | 92(73%) |
| | Others | 61(28.1%) | 62(30.1%) | 34(27%) |
| | p-value | 0.512 | 0.240 | 0.7 |
| Religion | Hindu | 212(97.7%) | 203(98.5%) | 123(97.6%) |
| | Others | 5(2.3%) | 3(1.5%) | 3(2.4%) |
| | p-value | 0.820 | 0.703 | 0.964 |
| Grade | Eleven | 104(47.9%) | 98(47.6%) | 58(46%) |
| | Twelve | 113(52.1%) | 108(52.4%) | 68(54%) |
| | p-value | 0.246 | 0.213 | 0.223 |
| Education system | Public | 85(39.2%) | 76(36.9%) | 24(19%) |
| | Private | 132(60.8%) | 130(63.1%) | 102(81%) |
| | p-value | **0.003***  | 0.089 | **<0.001*** |
| Current living condition | Living with family members | 163(75.1%) | 155(75.2%) | 98(77.8%) |
| | Others | 54(24.9%) | 51(24.8%) | 28(22.2%) |
| | p-value | 0.669 | 0.736 | 0.541 |
| Mother* education | No formal schooling | 51(23.7%) | 44(21.7%) | 23(18.3%) |
| | Formal schooling | 164(76.3%) | 159(78.3%) | 103(81.7%) |
| | p-value | 0.425 | 0.077 | **0.025*** |
| Mother* occupation | Employed | 78(36.3%) | 75(36.9%) | 46(36.5%) |
| | Unemployed | 137(63.7%) | 128(63.1%) | 80(63.5%) |
| | p-value | 0.372 | 0.255 | 0.546 |
| Father# education | No formal schooling | 18(8.5%) | 18(8.9%) | 8(6.4%) |
| | Formal schooling | 194(91.5%) | 184(91.1%) | 117(93.6%) |
| | p-value | 0.078 | 0.172 | **0.046*** |
| Father# occupation | Employed | 203(95.8%) | 196(97%) | 118(94.4%) |
| | Unemployed | 9(4.2%) | 6(3%) | 7(5.6%) |
| | p-value | 0.717 | 0.094 | 0.492 |
| Parent's§ chronic illness | Yes | 43(20%) | 37(18%) | 24(19%) |
| | No | 172(80%) | 168(82%) | 102(81%) |
| | p-value | 0.794 | 0.406 | 0.859 |
| Parent's§ smoking habit | Yes | 28(13%) | 26(12.7%) | 18(14.4%) |
| | No | 188(87%) | 179(87.3%) | 107(85.6%) |
| | p-value | 0.226 | 0.191 | 0.875 |
| Parent's§ smokeless tobacco use | Yes | 46(21.3%) | 54(26.3%) | 34(27.2%) |
| | No | 170(78.7%) | 151(73.7%) | 91(72.8%) |
| | p-value | 0.075 | 0.371 | 0.399 |
| Parent's§ drinking habit | Yes | 64(29.6%) | 55(26.8%) | 41(32.8%) |
| | No | 152(70.4%) | 150(73.2%) | 84(67.2%) |
| | p-value | 0.238 | 0.789 | 0.092 |

+n = 359 (two children did not disclose their gender) *n = 357 (four children do not have mother), #n = 350 (eleven children do not have father), §n = 358(three children do not have both of parents)

*Statistically significant(p < 0.05)

Percentages represent column% among those who reported the behaviour.

**Table 4. Association of refined vegetable oil use, current tobacco smoking, current alcohol use and current smokeless tobacco use (n = 361).**

| Characteristics | Categories | Refined vegetable oil user (yes = 70) n (%) | Current smoker(yes = 44) n (%) | Current alcohol user(yes = 37) n(%) | Current smokeless tobacco use(yes = 31) n(%) |
|---|---|---|---|---|---|
| Age | 16-17 | 34(48.6%) | 17(38.6%) | 11(29.7%) | 12(38.7%) |
| | 18-19 | 36(51.4%) | 27(61.4%) | 26(70.3%) | 19(61.3%) |
| | p-value | 0.904 | 0.188 | 0.019 | 0.283 |
| Gender⁺ | Male | 28(41.2%) | 38(86.4%) | 34(91.9%) | 28(90.3%) |
| | Female | 40(58.8.6%) | 6(13.6%) | 3(8.1%) | 3(9.7%) |
| | p-value | 0.061 | **<0.001***  | **<0.001***  | **<0.001***  |
| Ethnicity | Brahmin/Chhetri | 41(58.6%) | 24(54.5%) | 21(56.8%) | 22(71%) |
| | Others | 29(41.4%) | 20(45.5%) | 16(43.2%) | 9(29%) |
| | p-value | **0.014***  | **0.012***  | 0.05 | 0.966 |
| Religion | Hindu | 68(97.1%) | 42(95.5%) | 36(97.3%) | 30(96.8%) |
| | Others | 2(2.9%) | 2(4.5%) | 1(2.7%) | 1(3.2%) |
| | p-value | 0.890 | 0.450 | 1 | 1 |
| Grade | 11 | 38(54.3%) | 20(45.5%) | 14(37.8%) | 14(45.2%) |
| | 12 | 32(45.7%) | 24(54.5%) | 23(62.2%) | 17(54.8%) |
| | p-value | 0.659 | 0.482 | 0.106 | 0.541 |
| Education system | Public | 37(52.9%) | 18(40.9%) | 14(37.8%) | 10(32.3%) |
| | Private | 33(47.1%) | 26(59.1%) | 23(62.2%) | 21(67.7%) |
| | p-value | **0.001***  | 0.249 | 0.531 | 0.903 |
| Current living condition | Living with family members | 46(65.7%) | 32(72.7%) | 31(83.8%) | 24(77.4%) |
| | Others | 24(34.3%) | 12(27.3%) | 6(16.2%) | 7(22.6%) |
| | p-value | **0.026***  | 0.599 | 0.237 | 0.836 |
| Mother* education | No formal schooling | 21(30.4%) | 5(11.6%) | 6(16.2%) | 5(16.7%) |
| | Formal schooling | 48(69.6%) | 38(88.4%) | 31(83.8%) | 25(83.3%) |
| | p-value | 0.266 | **0.029***  | 0.183 | 0.260 |
| Mother* occupation | Employed | 26(37.7%) | 13(30.2%) | 10(27%) | 9(30%) |
| | Unemployed | 43(62.3%) | 30(69.8%) | 27(73%) | 21(70%) |
| | p-value | 0.530 | 0.535 | 0.315 | 0.592 |
| Father# Education | No formal schooling | 8(11.8%) | 5(11.6%) | 2(5.7%) | 2(6.7%) |
| | Formal schooling | 60(88.2%) | 38(88.4%) | 33(94.3%) | 28(93.3%) |
| | p-value | 0.789 | 1 | 0.457 | 0.642 |
| Father# Occupation | Employed | 65(95.6%) | 42(97.7%) | 34(97.1%) | 29(96.7%) |
| | Unemployed | 3(4.4%) | 1(2.3%) | 1(2.9%) | 1(3.3%) |
| | p-value | 1 | 0.717 | 0.932 | 1 |
| Parent's§ chronic illness | Yes | 20(29%) | 6(14%) | 8(21.6%) | 6(20%) |
| | No | 49(71%) | 37(86%) | 29(78.4%) | 24(80%) |
| | p-value | **0.026***  | 0.324 | 0.738 | 0.949 |
| Parent's§ smoking habit | Yes | 15(21.7%) | 13(30.2%) | 9(24.3%) | 5(16.7%) |
| | No | 54(78.3%) | 30(69.8%) | 28(75.7%) | 25(83.3%) |
| | p-value | 0.121 | **0.002***  | 0.085 | 0.764 |
| Parent's§ smokeless tobacco use | Yes | 21(30.4%) | 20(46.5%) | 15(40.5%) | 9(30%) |
| | No | 48(69.6%) | 23(53.5%) | 22(59.5%) | 21(70%) |
| | p-value | 0.217 | **<0.001***  | **0.017***  | 0.471 |
| Parent's§ alcohol use | Yes | 28(40.6%) | 16(37.2%) | 16(43.2%) | 12(40%) |
| | No | 41(59.4%) | 27(62.8%) | 21(56.8%) | 18(60%) |
| | p-value | **0.003***  | 0.123 | **0.022***  | 0.105 |

⁺*n = 359 (two children did not disclose their gender)*

**n = 357 (four children do not have mother),*

#*n = 350 (eleven children do not have father)*

§*n = 358(three children do not have both of parents)*

**Statistically significant(p < 0.05)*

Percentages represent column% among those who reported the behaviour.

**Table 5. Association and odds ratio of different predictors and independent variables.**

| Category | Predictor | B (SE) | p-value | UOR (95% CI) | AOR (95% CI) |
|---|---|---|---|---|---|
| Processed Food Consumption | Gender (Male vs. Female) | −0.516 (0.256) | **0.044**[*] | 0.686 [0.439, 1.072] | 0.597 [0.362, 0.986] |
| | Ethnicity (Brahmin/Chhetri vs. Others) | 0.775 (0.288) | **0.007**[*] | — | 2.171 [1.234, 3.821] |
| | Grade (12 vs. 11) | −0.602 (0.255) | 0.018 | 0.544 [0.348, 0.850] | 0.548 [0.332, 0.903] |
| | Mother's Education (Formal vs. No Formal Schooling) | 1.063 (0.354) | **0.003**[*] | 2.030 [1.164, 3.538] | 2.895 [1.447, 5.793] |
| | Age (19–18 vs. 16–17) | −0.046 (0.259) | 0.860 | 0.741 [0.477, 1.150] | 0.955 [0.575, 1.586] |
| Added Salt Intake | Grade (12 vs. 11) | −0.436 (0.324) | 0.178 | 0.666 [0.374, 1.186] | 0.646 [0.343, 1.219] |
| | Mother's Education (Formal vs. No Formal Schooling) | −0.585 (0.366) | 0.110 | 0.545 [0.297, 1.002] | 0.557 [0.272, 1.141] |
| Sedentary Behaviour | Education System (Private vs Public) | −0.696 (0.239) | **0.004**[*] | 0.499 [0.312, 0.796] | 0.351 [0.202, 0.609] |
| | Parent's Smokeless Tobacco Use (Yes vs. No) | −0.440 (0.248) | 0.076 | 0.644 [0.396, 1.047] | 0.510 [0.280, 0.930] |
| | Parent's Drinking Habit (Yes vs. No) | 0.291 (0.247) | 0.239 | 1.337 [0.825, 2.169] | 2.380 [1.275, 4.442] |
| Insufficient Fruit & Vegetable Intake | Grade (12 vs. 11) | 0.221 (0.239) | 0.355 | 1.304 [0.859, 1.980] | 1.247 [0.781, 1.991] |
| | Education System (Private vs Public) | −0.656 (0.265) | **0.013**[*] | 0.678 [0.433, 1.063] | 0.519 [0.309, 0.873] |
| | Mother's Education (Formal vs. No Formal Schooling) | 0.355 (0.295) | 0.229 | 1.539 [0.952, 2.488] | 1.426 [0.800, 2.544] |
| | Father's Education (Formal vs. No Formal Schooling) | 0.307 (0.394) | 0.436 | 1.597 [0.813, 3.139] | 1.359 [0.628, 2.942] |
| | Father's Occupation (Employed vs. Unemployed) | 0.895 (0.546) | 0.101 | 2.367 [0.841, 6.666] | 2.447 [0.840, 7.129] |
| | Parent's Smoking Habit (Yes vs. No) | −0.542 (0.374) | 0.147 | 0.678 [0.378, 1.216] | 0.582 [0.279, 1.211] |
| Physical Inactivity | Age (19–18 vs. 16–17) | 0.200 (0.259) | 0.439 | 1.368 [0.885, 2.115] | 1.222 [0.736, 2.029] |
| | Gender (Male vs. Female) | 0.114 (0.248) | 0.646 | 1.304 [0.843, 2.017] | 1.121 [0.690, 1.822] |
| | Grade (12 vs. 11) | 0.045 (0.249) | 0.856 | 1.310 [0.849, 2.021] | 1.046 [0.642, 1.704] |
| | Education System (Private vs Public) | 1.033 (0.295) | **<0.001**[*] | 2.935 [1.754, 4.913] | 2.810 [1.574, 5.013] |
| | Mother's Education (Formal vs. No Formal Schooling) | 0.247 (0.320) | 0.441 | 1.830 [1.073, 3.120] | 1.280 [0.683, 2.397] |
| | Father's Education (Formal vs. No Formal Schooling) | 0.467 (0.461) | 0.311 | 2.250 [0.998, 5.072] | 1.596 [0.647, 3.938] |
| | Parent's Drinking Habit (Yes vs. No) | 0.371 (0.302) | 0.219 | 1.507 [0.934, 2.431] | 1.449 [0.802, 2.619] |
| Refined Vegetable Oil Use | Sex (Male vs. Female) | −0.277 (0.311) | 0.374 | 0.798 [0.467, 1.362] | 0.758 [0.412, 1.395] |
| | Ethnicity (Brahmin/Chhetri vs. Others) | −0.320 (0.313) | 0.307 | 0.509 [0.296, 0.875] | 0.726 [0.394, 1.341] |
| | Education System (Private vs Public) | −1.124 (0.319) | **<0.001**[*] | 0.356 [0.209, 0.607] | 0.325 [0.174, 0.608] |
| | Parent's Chronic Illness (Yes vs. No) | 0.751 (0.341) | **0.028**[*] | 1.951 [1.068, 3.565] | 2.119 [1.086, 4.133] |
| | Parent's Alcohol Use (Yes vs. No) | 1.004 (0.365) | **0.006**[*] | 2.137 [1.232, 3.706] | 2.728 [1.333, 5.584] |
| Current Smoker | Sex (Male vs. Female) | 2.428 (0.507) | **<0.001**[*] | 9.500 [3.901, 23.134] | 11.336 [4.196, 30.623] |
| | Ethnicity (Brahmin/Chhetri vs. Others) | −0.732 (0.425) | 0.085 | 0.447 [0.235, 0.850] | 0.481 [0.209, 1.105] |
| | Mother's Education (Formal vs. No Formal Schooling) | 1.542 (0.612) | **0.012**[*] | 2.821 [1.075, 7.405] | 4.675 [1.408, 15.523] |
| | Parent's Smoking Habit (Yes vs. No) | 1.117 (0.551) | **0.043**[*] | 2.979 [1.435, 6.185] | 3.055 [1.037, 8.997] |
| | Parent's Smokeless Tobacco Use (Yes vs. No) | 1.010 (0.438) | **0.021**[*] | 3.159 [1.638, 6.090] | 2.745 [1.163, 6.479] |
| | Parent's Drinking Habit (Yes vs. No) | −0.415 (0.478) | 0.385 | 1.684 [0.864, 3.283] | 0.661 [0.259, 1.684] |
| Alcohol Use | Age (19–18 vs. 16–17) | 0.557 (0.469) | 0.235 | 2.364 [1.130, 4.943] | 1.745 [0.696, 4.377] |
| | Sex (Male vs. Female) | 2.647 (0.641) | **<0.001**[*] | 16.738 [5.035, 55.644] | 14.112 [4.015, 49.600] |
| | Ethnicity (Brahmin/Chhetri vs. Others) | −0.781 (0.453) | 0.085 | 0.505 [0.252, 1.011] | 0.458 [0.188, 1.113] |
| | Parent's Smokeless Tobacco Use (Yes vs. No) | 0.528 (0.465) | 0.256 | 2.316 [1.143, 4.694] | 1.695 [0.682, 4.212] |
| | Parent's Alcohol Use (Yes vs. No) | 0.237 (0.461) | 0.607 | 2.221 [1.106, 4.459] | 1.268 [0.513, 3.133] |

*(Continued)*

## PLOS One

**Table 5.** (Continued)

| Category | Predictor | B (SE) | p-value | UOR (95% CI) | AOR (95% CI) |
|---|---|---|---|---|---|
| Smokeless Tobacco Use | Sex (Male vs. Female) | 2.557 (0.646) | **<0.001**\* | 13.176 [3.926, 44.221] | 12.896 [3.636, 45.740] |
| | Parent's Smoking Habit (Yes vs. No) | –0.132 (0.661) | 0.842 | 1.167 [0.426, 3.196] | 0.877 [0.240, 3.202] |
| | Parent's Smokeless Tobacco use (Yes vs. No) | 0.103 (0.509) | 0.840 | 1.351 [0.594, 3.070] | 1.108 [0.409, 3.005] |
| | Parent's Alcohol Use (Yes vs. No) | 0.415 (0.472) | 0.379 | 1.876 [0.868, 4.055] | 1.515 [0.600, 3.825] |

\***Statistically significant (p < 0.05)**

Similarly smokeless tobacco use (OR=12.896) and current alcohol use (OR=14.112) were higher among males, reflecting societal norms in Nepal that accept tobacco and alcohol consumption among men. The lack of effective enforcement of tobacco control measures likely contributes to easy access to tobacco products for adolescents [18]. The widespread availability of alcohol in the local market may also account for the higher rates observed in this study. Gender and cultural norms likely also play a significant role in shaping these behaviours in the context of Nepal.

## Strengths and limitations

Our study used a standardized questionnaire designed for adolescents to estimate and examine the relationship between key factors associated with CVD risk. To our knowledge, these factors have not been systematically explored in previous studies in Nepal.

Several factors may have influenced the findings of this study. First, the results represent adolescents aged 16–19 years from Tulsipur Sub-Metropolitan City and may not be generalizable to adolescents in other regions of Nepal, particularly rural areas. Although stratification was conducted by public and private schools, it was not extended to rural versus urban schools. However, Tulsipur, located in the inner terai region, includes both terai and hilly characteristics, and its rapid urbanization reflects a mix of urban and rural settings. Given the adequate sample size, the findings may may still provide some insights applicable to similar diverse (urban, rural, hilly, and plain) landscapes in other settings.

The use of a self-administered questionnaire may have introduced self-report and recall bias Students might misinterpret information due to less knowledge, personal perception and inaccurately recalling past events. To minimize these risks, we used standardized questions with clear wording and provided clarification before data collection. We also reduced the possibility of information bias in group settings by limiting the number of participants to 15 per group and ensuring adequate spacing to minimize peer influence.

## Conclusion

This study provides evidence that behavioural risk factors for CVD are disproportionately distributed among specific subgroups of adolescents, including males, and those with a parental history of CVD or lower parental education. The high prevalence of these risk factors during adolescence may lead to an increased incidence of CVD in adulthood, potentially resulting in significant health and economic burdens. To address this, targeted interventions may include age-appropriate school health education on cardiovascular risk factors integrated into the curriculum; promotion of daily physical activity; restrictions on processed fast foods and sugary beverages in school cafeterias; regular health screenings (e.g., blood pressure, BMI, blood glucose); mental health workshops and counseling sessions; and referral of high-risk children to health professionals.

Additionally, there must be improved monitoring and enforcement of existing policies, such as restrictions on smoking and alcohol use among adolescents and regulations discouraging the marketing of fast food, processed foods, and high-calorie drinks at the local government level.

## Supporting information

**S1 Appendix. Operational definitions.**
(DOCX)

**S2 Appendix. Global research questionnaire.**
(DOCX)

**S1 Table. Regression analysis.**
(DOCX)

## Author contributions

**Conceptualization:** Sita Bista, Bishow Puri, Poshan Thapa, Buna Bhandari.

**Data curation:** Sita Bista, Bishow Puri.

**Formal analysis:** Sita Bista, Bishow Puri, Sanju Maharjan.

**Investigation:** Sita Bista, Bishow Puri, Buna Bhandari.

**Methodology:** Sita Bista, Bishow Puri, Sanju Maharjan, Buna Bhandari.

**Resources:** Sita Bista.

**Supervision:** Poshan Thapa, Buna Bhandari.

**Validation:** Bishow Puri, Buna Bhandari.

**Visualization:** Sita Bista, Bishow Puri, Sanju Maharjan, Buna Bhandari.

**Writing – original draft:** Sita Bista, Bishow Puri, Sanju Maharjan.

**Writing – review & editing:** Sita Bista, Bishow Puri, Sanju Maharjan, Poshan Thapa, Buna Bhandari.

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
