## [Decision Letter · Decision Letter 0]

8 Dec 2024

PONE-D-24-49797Behavioural risk factors for cardiovascular disease among adolescents of secondary level school in a sub-metropolitan setting in Nepal: A cross-sectional studyPLOS ONE

Dear Dr. Bista,

Thank you for submitting your manuscript to PLOS ONE. After careful consideration, we feel that it has merit but does not fully meet PLOS ONE’s publication criteria as it currently stands. Therefore, we invite you to submit a revised version of the manuscript that addresses the points raised during the review process.

We look forward to receiving your revised manuscript.

Kind regards,

Shalik Ram Dhital, PhD

Academic Editor

PLOS ONE

Additional Editor Comments:

Please address two reviewers comments  satisfactorily.  I will look your paper after addressing reviewers  comments. 

Please go through each comments of each reviewers and revise your manuscript according to PLOS ONE submission guideline. 

Reviewers' comments:

Reviewer's Responses to Questions

**Comments to the Author**

1. Is the manuscript technically sound, and do the data support the conclusions?

Reviewer #1: Partly

Reviewer #2: No

2. Has the statistical analysis been performed appropriately and rigorously? 

Reviewer #1: N/A

Reviewer #2: No

3. Have the authors made all data underlying the findings in their manuscript fully available?

Reviewer #1: Yes

Reviewer #2: Yes

4. Is the manuscript presented in an intelligible fashion and written in standard English?

Reviewer #1: Yes

Reviewer #2: Yes

5. Review Comments to the Author

Reviewer #1: Review of "Behavioural risk factors for cardiovascular disease among adolescents in Nepal: A cross-sectional study"

Introduction and Objectives: The manuscript aims to assess the prevalence of cardiovascular disease (CVD) risk factors among adolescents in a semi-urban setting in Nepal. This objective is significant, as early exposure to risk factors can exacerbate the burden of CVD later in life, especially in resource-limited settings.

Methodology: This study utilizes a cross-sectional design with a sample of 361 adolescents aged 16 to 19, selected via stratified proportionate sampling. The methods are appropriate for descriptive epidemiology; however, several issues limit the rigor of the analysis:

- Sampling Bias: Stratification only by school type (public/private) may inadequately represent other socio-economic or demographic factors relevant to CVD risk, such as family income and geographic region.

- Self-Reporting: Data were gathered through self-administered questionnaires, which can introduce recall and reporting bias, particularly for socially influenced behaviors (e.g., smoking and alcohol consumption). Mitigating steps like supervision were attempted, yet objective measures (such as biomarkers for smoking) could strengthen validity.

- Statistical Analysis: The study employs both univariate and bivariate analyses to investigate associations between CVD risk factors and demographic variables. While adequate, the study lacks multivariate analysis, which could better control for potential confounding factors, improving the robustness of findings on associations between behaviors and socio-demographic variables.

Results and Data Presentation: Results are comprehensive, with clear prevalence statistics for risk factors such as calorie-dense drink consumption, sedentary behavior, and smoking. However, the presentation would benefit from:

- Further Detailing: Additional tables or visual aids summarizing complex associations (e.g., those affected by parental influence) would enhance clarity.

- Statistical Interpretation: The manuscript lacks sufficient emphasis on confidence intervals or effect sizes, making it challenging to gauge the strength of associations. This omission reduces interpretive depth, particularly in relation to public health interventions.

Utility and Novelty of Findings: The study reveals high prevalence rates of CVD risk behaviors, highlighting critical areas for intervention, including physical inactivity and poor dietary habits. While the study addresses a vital public health issue, the lack of innovative or highly detailed insights may limit its impact for top-tier journals. Most findings, such as associations between adolescent smoking and parental behaviors, align closely with existing literature from similar settings. Nonetheless, the study’s context (Nepal) may be valuable for policymakers within the region.

Recommendations for Revision:

1. Enhance Methodological Rigor: Addressing self-reporting limitations with more objective measures, if feasible, or by discussing potential biases in-depth, could enhance the study's reliability.

2. Statistical Refinement: Adding multivariate analyses would help substantiate causal inferences and control for confounding variables, lending greater credibility to findings.

3. Data Visualization: Incorporate more visual data representations, particularly for associations impacted by socio-demographic factors, to improve readability and impact.

Conclusion: The manuscript provides a well-structured but relatively standard analysis of behavioral CVD risk factors in a semi-urban Nepali adolescent population. Although the findings reinforce global public health recommendations, methodological limitations (e.g., reliance on self-report, lack of multivariate analysis) and the narrow focus on descriptive statistics reduce the study’s potential impact.

Reviewer #2: Comments

Thank you for providing me opportunity to review the paper. I've some comments listed below:

1. In the title sub-metropolitan setting is misleading as this study is conducted in Tulsipur sub-metropolitan city only. Abstract doesn't says what is the study population is?

2. Does your study really used simple random lottery method to select students from the school. IF so how have you reach out the specific students to collect information? If your study is stratified proportionate sampling, you should present the results stratified by rural and urban schools and further stratified by 11 and 12th grade. You need to be cautious while merging two groups and estimate the proportion for all population.

Please add details on how you reach out to the students.

3. Line 79: Missing inline citation

3. Line number 94-96: Who or how many people collected data and who supervised whom? It needs to be clarified.

4. Line 103: what are you calculating confidence interval of ??

5. For those with age less than 18 what have you done.

5. Statistical analysis part misses about SD that you calculated in results

6. Line 117: Please keep all details of participants characteristics within manuscript.

7. Calculate % and its 95% CI rather than just point estimation (use Wilson method or other appropriate method to calculate 95% CI)

9. Results from supplementary tables are in the discussion. Please keep whatever you discuss in the main text

10. Include number of missing data each variables if missing.

11. Reference should be formatted as per journal requirement

12. requires extensive grammar correction

6. PLOS authors have the option to publish the peer review history of their article (what does this mean? ). If published, this will include your full peer review and any attached files.

**Do you want your identity to be public for this peer review?** For information about this choice, including consent withdrawal, please see our Privacy Policy .

Reviewer #1: No

Reviewer #2: No

---

## [Author Response · Author response to Decision Letter 1]

2 Mar 2025

Dear Dr. Dhital,

Academic editor, PLOS One

We thank you and the reviewers for the constructive comments on our manuscript, “Behavioural risk factors for cardiovascular disease among adolescents of secondary level school in a sub-metropolitan setting in Nepal: A cross-sectional study (PONE-D-24-49797).” We have carefully addressed all comments and revised the manuscript accordingly. Below is a point-by-point response to each comment.

Recommendations for Revision from Reviewer 1 :

1. Enhance Methodological Rigor: Addressing self-reporting limitations with more objective measures, if feasible, or by discussing potential biases in-depth, could enhance the study's reliability.

Response: Thank you for your insightful comments. We have revised the limitation section accordingly and added the potential bais in the revised manuscript. Changes can be seen in page no 20, line no 243-249.

2. Statistical Refinement: Adding multivariate analyses would help substantiate causal inferences and control for confounding variables, lending greater credibility to findings.

Response: We have incorporated the reviewer’s suggestion and conducted the multivariate analysis (Binary logistic regression) along with the Chi square test to ensure the robustness of association. The multivariate analysis data analysis is presented in table 5and findings are updated throughout the manuscript in the result ( page 16-17 and Line no 170-178) and discussion as well.

3. Data Visualization: Incorporate more visual data representations, particularly for associations impacted by socio-demographic factors, to improve readability and impact

Response: Thanks you for the suggestion. We have substituted table 1 with the bar chart displaying the prevalence of behavioral risk factors in adolescents and binary logistic regression table 5) was added to results section displaying significance of association and Odds ratio.

Reviewer #2: Comments

1. In the title sub-metropolitan setting is misleading as this study is conducted in Tulsipur sub-metropolitan city only. Abstract doesn't says what is the study population is?

Response: Thank you for your valuable remarks. We have revised the title to specifically mention "Tulsipur Sub-Metropolitan City" instead of the general term "sub-metropolitan setting" for clarity. We have also clarified the study population in the abstract ( line 17-18)

2. Does your study really used simple random lottery method to select students from the school. If so how have you reach out the specific students to collect information? If your study is stratified proportionate sampling, you should present the results stratified by rural and urban schools and further stratified by 11 and 12th grade. You need to be cautious while merging two groups and estimate the proportion for all population.

Please add details on how you reach out to the students.

Response: Thank you for providing opportuntiy to clarify our sampling approach. A stratified proportionate sampling method was employed using sampling frame obtained from municipal education section. The two strata were defined based on schools’ affiliation (Public/Private), with the sample size proportionally distributed based on the student population in each stratum. Two public and two private secondary schools were randomly selected. The students were divided into grades 11 and 12, with a proportional distribution according to age. Within each school, students were selected through a simple random lottery method using a random number generator. The student’s records and details of each selected school were obtained from the school administration. Each student were assigned a unique identification number. Based on proportionate sample of selected schools, Microsoft excel was used to generate a predefined and non-repeating number. The randomly generated numbers were cross-verified with the student details. Then after, selected students were invited to participate in study. If selected students didn’t provide consent or were absent, alternative students were taken through additional random numbers generated. These information are clarified in the revised manuscript (page no 4-5, line no 86-98)

There was stratification by public and private schools, then proportionate sampling was taken from 11th and 12th grade but not stratified by rural and urban schools and which is included as a limitation of the study in the limitation section.

3. Line 79: Missing inline citation

Response: We have inserted citation.

4. Line number 94-96: Who or how many people collected data and who supervised whom? It needs to be clarified.

Respone: The data was collected by the first and second author (SB and BP) who were supervised by the senior author (BB and PT). We have calrified our approach in the revised manuscript.

5. Line 103: what are you calculating confidence interval of ??

Response: The 95% confidence interval was calculated to measure the precision of the estimated associations in the chi-square test.

6. For those with age less than 18 what have you done.

Response: Both written parental consent and student’s consent were taken for age less than 18 and those students with aged 18 or above provided their written consent. On the first day, Parents were provided with parental consent form through students, then only if agreed by parents for involvement in study, students were provided with written consent form. This has been clarified in the revised manuscript ( page 6, line 117-121)

7. Statistical analysis part misses about SD that you calculated in results

Response: Thank you. This issue has been addressed at data processing and analysis section. ( page no 7, line no 127)

8. Line 117: Please keep all details of participants characteristics within manuscript.

Response: We have moved the table of characteristics of participants from the supplementary file to the manuscript (Table 1page 7, line 136).

9. Calculate % and its 95% CI rather than just point estimation (use Wilson method or other appropriate method to calculate 95% CI)

10. Response: Adjusted Odds ratio has been calculated by using binary logistic regression analysis.

Results from supplementary tables are in the discussion. Please keep whatever you discuss in the main text

Response: Data discussed in the manuscript are presented within the manuscript; only further information is provided at the supplementary file section.

11. Include number of missing data each variables if missing.

Response: Thank you for your suggestion to include number of missing data each variables if missing. We didn’t find any missing values of variables after carefully reviewing the dataset, except for the parent’s characteristics that has been clarified in the footnote of Table 1.

12. Reference should be formatted as per journal requirement

Response: Thank you, we have formatted as per journal requirement

13. requires extensive grammar correction

Response: Thank you. We have revised and proof read the manuscript for the language and grammatical errors.

We sincerely appreciate the opportunity to revise and resubmit our manuscript. We are confident that the quality of the revised manuscript has significantly improved and now meets the expectations of the reviewers and the journal.

Sincerely,

Sita Bista

---

## [Decision Letter · Decision Letter 1]

28 Apr 2025

PONE-D-24-49797R1Behavioural risk factors for cardiovascular disease among adolescents of secondary school in Tulsipur sub-metropolitan city, Nepal: A cross-sectional studyPLOS ONE

Dear Dr. Bista,

Thank you for submitting your manuscript to PLOS ONE. After careful consideration, we feel that it has merit but does not fully meet PLOS ONE’s publication criteria as it currently stands. Therefore, we invite you to submit a revised version of the manuscript that addresses the points raised during the review process.

We look forward to receiving your revised manuscript.

Kind regards,

Chiranjivi Adhikari, MPH, MHEd., PhD Candidate

Academic Editor

PLOS ONE

Additional Editor Comments:

Dear previous Editors, and all the reviewers,

I would also take an opportunity to thank our esteemed reviewers, and previous editors, for their scientific comments.

Dear Authors,

Thank you for submitting the scientific work "Behavioral risk factors for cardiovascular disease among adolescents of secondary school in Tulsipur sub-metropolitan city, Nepal: A cross-sectional study". The manuscript is well documented in language and coherence, clarification and revisions for the following comments, along with those submitted by the previous editors, and reviewers, are requested though:

1.In line 82-85, N=5262, is this total of school going adolsescent of Tulsipur, or of only 16-19? pls clarify, and adjust for sampling accordingly.

2. Line 104-105, What was the pretested sample population, and place, pls specify.

3. Table 2, 3, 4; percent total was observed, but columwise p-value does not comply with, so columwise % total may be better interpretable?!

4. Table 5 shows the UOR (?), and the list of regressors are too many and poorly interpretable, such as mother's education and ethnicity may be just the confounders for processed food consumption. So better to reduce the risk factors with further adjustment with modelling and so, better interpretatble to discuss more contextually. And then, re-discuss as of the new results.

With regards,

Chiranjivi

AE

Reviewers' comments:

Reviewer's Responses to Questions

**Comments to the Author**

1. If the authors have adequately addressed your comments raised in a previous round of review and you feel that this manuscript is now acceptable for publication, you may indicate that here to bypass the “Comments to the Author” section, enter your conflict of interest statement in the “Confidential to Editor” section, and submit your "Accept" recommendation.

Reviewer #1: All comments have been addressed

Reviewer #3: (No Response)

2. Is the manuscript technically sound, and do the data support the conclusions?

Reviewer #1: Yes

Reviewer #3: (No Response)

3. Has the statistical analysis been performed appropriately and rigorously? 

Reviewer #1: Yes

Reviewer #3: (No Response)

4. Have the authors made all data underlying the findings in their manuscript fully available?

Reviewer #1: (No Response)

Reviewer #3: (No Response)

5. Is the manuscript presented in an intelligible fashion and written in standard English?

Reviewer #1: (No Response)

Reviewer #3: (No Response)

6. Review Comments to the Author

Reviewer #1: All comments have been addressed. I do think the manuscript has been properly improved and can be accepted.

Reviewer #3: Title and Abstract

Comment:

1. The phrase “affecting one-third of adolescents” in the abstract may be misleading. Please clarify whether this refers to actual/clinical CVD or its risk factors. (Same comment for introduction part; Line no 43)

2. “Schoolswere” has a typo

3. Spacing and formatting inconsistent on Key words section (It applies to the whole manuscript).

Introduction

Comment:

1. The paragraph shifts abruptly from global data to adolescent.

Suggestion: It would be easier to understand if the data flowed from global to regional to national and local.

Method

1. What was the mechanism of identifying students with a history of CVDs or other chronic diseases (any students found during data collection if yes how)? Please mentioned it.

2. The manuscript mentions the assessment of cardiovascular disease (CVD) risk behaviors among adolescents. The authors need to clarify whether any procedures were in place to ethically manage participants who were identified as being at heightened risk during the study.

Result

1. author used asterisks (*) at multiples places, used different indicators (eg #) with clear footnotes and explanations (it applies to the other sections of results eg table no 3, 4)

2. Spelling error (line no 146 )

3. In the “Association Between CVD Risk Factors and Socio-demographic Characteristics” section, the author needs to indicate the significant result within the table by using an indicator (e.g., 0.028*) and highlighting the result; it could help the reader.

Discussion

1. In the first paragraph of the discussion, the author had summarized the findings, which can be more concise, and if the author put “what is going to be discussed,” it would be easier for readers. Then only start the discussion in another paragraph.

2. The author discussed it well, but it would be better to discuss separately to risks. For eg discuss physical activity in one single, calorie intake in another.

Conclusion/Recommendation

1. The word used by author “targeted intervention” sounds vague, what types of interventions and to whom/who are the high risk group? Could be mentioned, and it will be more specific.

7. PLOS authors have the option to publish the peer review history of their article (what does this mean? ). If published, this will include your full peer review and any attached files.

**Do you want your identity to be public for this peer review?** For information about this choice, including consent withdrawal, please see our Privacy Policy .

Reviewer #1: No

Reviewer #3: No

---

## [Author Response · Author response to Decision Letter 2]

7 Aug 2025

12th July, 2025

Dear Dr. Adhikari,

Academic editor, PLOS One

We thank you and the reviewers for the constructive comments on our manuscript, “Behavioural risk factors for cardiovascular disease among adolescents of secondary level school in a sub-metropolitan setting in Nepal: A cross-sectional study (PONE-D-24-49797).” We have carefully addressed all comments and revised the manuscript accordingly. Below is a point-by-point response to each comment.

Recommendations for Revision

Academic Editor

1.In line 82-85, N=5262, is this total of school going adolescent of Tulsipur, or of only 16-19? Response: Thank you for your comment. This is the total population of secondary level school adolescents of age between 16-19 studying in only grade 11 and 12, which is clarified in the manuscript( page 5, line 81)

2. Line 104-105, What was the pretested sample population, and place, pls specify.

Response: Thank. We have provided details of the pretesting in the revised manuscript as follows

Pretesting was conducted among 37 students from Ghorahi sub-metropolitan city, a location with similar characteristics to the study area.(page 6 , line 103-105). The pre-testing data was not included in final analysis.

3. Table 2, 3, 4; percent total was observed, but columwise p-value does not comply with, so columwise % total may be better interpretable?!

Response: Thank you for your suggestion. We have updated all tables with the column wise percentage and total as evidenced in revised table Table 2, 3 and 4.

4. Table 5 shows the UOR (?), and the list of regressors are too many and poorly interpretable, such as mother's education and ethnicity may be just the confounders for processed food consumption. So better to reduce the risk factors with further adjustment with modelling and so, better interpretable to discuss more contextually. And then, re-discuss as of the new results.

Response: Thank you for your suggestion. We have conducted further analysis adjusting for potential confounders using a multivariable logistic model and now present both Adjusted Odds Ratios (AORs) and Unadjusted Odds Ratios (UORs) in the revised Table 5 (page 21, line no 181). The discussion section has also been updated to reflect and contextualize these updated findings accordingly.

Reviewer #3:

Title and Abstract

Comment:

1. The phrase “affecting one-third of adolescents” in the abstract may be misleading. Please clarify whether this refers to actual/clinical CVD or its risk factors. (Same comment for introduction part; Line no 43)

Response: Thank you for your valuable remarks. We acknowledge that the original phrasing may have caused confusion. The statement was intended to refer to cardiovascular disease (CVD) affecting one-third of adult population. We have revised the sentence in both the abstract (Line no.13, page no.2) and the introduction (Line no 40, page no. 3) to clarify this confusion.

2. “Schoolswere” has a typo

Response: Thank you. We have edited it. (Line no 18, page no.2)

3. Spacing and formatting inconsistent on Key words section (It applies to the whole manuscript).

Response: We appreciate for highlighting the formatting issue. We have carefully reviewed and formatted the whole manuscript including spacing of the Keywords section.

Introduction

Comment:

1. The paragraph shifts abruptly from global data to adolescent.

Suggestion: It would be easier to understand if the data flowed from global to regional to national and local.

Response: Thank you for the insightful suggestion. We have revised the paragraph to enhance clarity and improve the flow, presenting the data in a logical sequence from global to regional to national levels, while maintaining a focus on the global burden and its relevance to adolescents. Unfortunately, sufficient data on the local burden are not available. (see changes page 3, line 38-46)

Method

1. What was the mechanism of identifying students with a history of CVDs or other chronic diseases (any students found during data collection if yes how)? Please mentioned it.

Response: Thank you for your valuable comment. All data were based on the students’ self-reported responses regarding their history of cardiovascular disease (CVD). None of the participants reported having a history of CVDs or other chronic conditions at the time of data collection.

2. The manuscript mentions the assessment of cardiovascular disease (CVD) risk behaviors among adolescents. The authors need to clarify whether any procedures were in place to ethically manage participants who were identified as being at heightened risk during the study.

Response: Thank you for this important observation. While our study did not involve clinical assessment or diagnosis of individual cardiovascular risk, we acknowledge the ethical responsibility when identifying high-risk behaviours among adolescents. To address this, we shared the de-identified and aggregated findings with the relevant municipal and school authorities to raise awareness about the high prevalence of major CVD risk factors and the importance of school-based health education program. As noted in the conclusion, this highlights an area for future research and opportunity for school based intervention. However, due to limited resources and observational nature of the study, we were not able to offer individual counseling or follow-up for participants identified as having potentially high CVD risk behaviors.

Result

1. author used asterisks (*) at multiples places, used different indicators (eg #) with clear footnotes and explanations (it applies to the other sections of results eg table no 3, 4)

Response: We appreciate your suggestions. We have revised the use of footnote indicators across all applicant tables, including Tables 1, 2, 3 and 4 to enhance clarity and maintain consistency throughout the manuscript.

2. Spelling error (line no 146)

Response: Thank you. We have corrected the errors on line 148, page no.12 and proofread the revised manuscript

3. In the “Association between CVD Risk Factors and Socio-demographic Characteristics” section, the author needs to indicate the significant result within the table by using an indicator (e.g., 0.028*) and highlighting the result; it could help the reader.

Response: Thank you for your feedback. We have highlighted the significant result consistently across all tables using an indicator.

Discussion

1. In the first paragraph of the discussion, the author had summarized the findings, which can be more concise, and if the author put “what is going to be discussed,” it would be easier for readers. Then only start the discussion in another paragraph.

Response: Thank you. We have revised our discussion accordingly in the revised manuscript including breaking a paragraph in line no 190-194, page no. 25.

2. The author discussed it well, but it would be better to discuss separately to risks. For eg discuss physical activity in one single, calorie intake in another.

Response: Thank you. We have discussed the individual risk factors in individual paragraphs.

Conclusion/Recommendation

1. The word used by author “targeted intervention” sounds vague, what types of interventions and to whom/who are the high risk group? Could it be mentioned, and it will be more specific.

Response: We have clarified the possible targeted intervention in the revised manuscript as follows:

Targeted interventions may include age-appropriate school health education on cardiovascular risk factors integrated into the curriculum; promotion of daily physical activity; restrictions on processed fast foods and sugary beverages in school cafeterias; regular health screenings (e.g., blood pressure, BMI, blood glucose); mental health workshops and counseling sessions; and referral of high-risk children to health professionals. (line 272-277, page 29)

We sincerely appreciate the opportunity to revise and resubmit our manuscript. We are confident that the quality of the revised manuscript has significantly improved and now meets the expectations of the reviewers and the journal.

Sincerely,

Sita Bista

---

## [Editor Report · Decision Letter 2]

17 Aug 2025

Behavioural risk factors for cardiovascular disease among adolescents of secondary school in Tulsipur sub-metropolitan city, Nepal: A cross-sectional study

PONE-D-24-49797R2

Dear Dr Bista,

We’re pleased to inform you that your manuscript has been judged scientifically suitable for publication and will be formally accepted for publication once it meets all outstanding technical requirements.

Kind regards,

Chiranjivi Adhikari, MPH, MHEd., PhD Candidate

Academic Editor

PLOS ONE
---

## [Editor Report · Acceptance letter]

PONE-D-24-49797R2

PLOS ONE

Dear Dr. Bista,

I'm pleased to inform you that your manuscript has been deemed suitable for publication in PLOS ONE. Congratulations! Your manuscript is now being handed over to our production team.

Kind regards,

on behalf of

Mr. Chiranjivi Adhikari

Academic Editor

PLOS ONE